# HCV Cascade of Care in HIV/HCV Co-Infected Individuals: Missed Opportunities for Micro-Elimination

**DOI:** 10.3390/v16060885

**Published:** 2024-05-30

**Authors:** Christos Thomadakis, Dimitrios Basoulis, Olga Tsachouridou, Konstantinos Protopapas, Vasilios Paparizos, Myrto Astriti, Maria Chini, Georgios Chrysos, Markos Marangos, Periklis Panagopoulos, Diamantis Kofteridis, Helen Sambatakou, Elpida Mastrogianni, Nikos Panatzis, Evmorfia Pechlivanidou, Mina Psichοgiou, Giota Touloumi

**Affiliations:** 1Department of Hygiene, Epidemiology and Medical Statistics, Medical School, National and Kapodistrian University of Athens, 115 27 Athens, Greece; cthomadak@med.uoa.gr (C.T.); npantaz@med.uoa.gr (N.P.); evmorfia.pechlivanidou@gmail.com (E.P.); 21st Department of Internal Medicine, Medical School, National and Kapodistrian University of Athens, 115 27 Athens, Greece; dimitris.bassoulis@gmail.com (D.B.); elpidamastrogianni@gmail.com (E.M.); mpsichog@yahoo.gr (M.P.); 3Infectious Diseases Unit, 1st Internal Medicine Department, AHEPA University Hospital, Aristotle University of Thessaloniki, 546 36 Thessaloniki, Greece; olgat_med@hotmail.com; 44th Department of Internal Medicine, Medical School, National and Kapodistrian University of Athens, Attikon University General Hospital, 124 62 Athens, Greece; kprotopapas@hotmail.com; 5AIDS Unit, Clinic of Venereologic & Dermatologic Diseases, Medical School, Syngros Hospital, National and Kapodistrian University of Athens, 161 21 Athens, Greece; vpaparizos@yahoo.gr; 61st Department of Internal Medicine and Infectious Diseases Unit, General Hospital of Athens G. Gennimatas, 115 27 Athens, Greece; myrto_astriti@hotmail.com; 73rd Department of Internal Medicine Infectious Diseases Unit, Red Cross General Hospital, 115 26 Athens, Greece; mariachini@gmail.com; 8Infectious Diseases Unit, Tzaneion General Hospital of Piraeus, 185 36 Athens, Greece; gchrysos@gmail.com; 9Department of Internal Medicine & Infectious Diseases, Patras University General Hospital, 265 04 Patras, Greece; mmarangos@yahoo.com; 10Infectious Diseases Unit, 2nd University Department of Internal Medicine, University Hospital of Alexandroupolis, Democritus University of Thrace, 681 00 Alexandroupolis, Greece; ppanago@med.duth.gr; 11Department of Internal Medicine, University Hospital of Heraklion, 715 00 Heraklion, Greece; kofterid@med.uoc.gr; 12HIV Unit, 2nd Department of Internal Medicine, National and Kapodistrian University of Athens, Medical School, Hippokration University General Hospital, 115 27 Athens, Greece; helensambatakou@msn.com

**Keywords:** HIV/HCV co-infection, DAAs, cascade of care (COC), HCV micro-elimination, missed opportunities

## Abstract

People living with HIV-HCV co-infection comprise a target group for HCV-micro-elimination. We conducted an HCV cascade of care (CoC) for HIV-HCV co-infected individuals living in Greece and investigated factors associated with different HCV-CoC stages. We analyzed data from 1213 participants from the Athens Multicenter AIDS Cohort Study. A seven-stage CoC, overall and by subgroup (people who inject drugs (PWID), men having sex with men (MSM), men having sex with women (MSW), and migrants], was constructed, spanning from HCV diagnosis to sustained virologic response (SVR). Logistic/Cox regression models were employed to identify factors associated with passing through each CoC step. Among 1213 anti-HCV-positive individuals, 9.2% died before direct-acting antiviral (DAA) availability. PWID exhibited higher mortality rates than MSM. Of 1101 survivors, 72.2% remained in care and underwent HCV-RNA testing. Migrants and PWID showed the lowest retention rates. HCV-RNA was available for 79.2% of those in care, with 77.8% diagnosed with chronic HCV. Subsequently, 71% initiated DAAs, with individuals with very low CD4 counts (<100 cells/μL) exhibiting lower odds of DAA initiation. SVR testing was available for 203 individuals, with 85.7% achieving SVR. The SVR rates did not differ across risk groups. In 2023, significant gaps and between-group differences persisted in HCV-CoC among HIV-HCV co-infected individuals in Greece.

## 1. Introduction

The World Health Organization estimates that the global burden of HIV infection is 39 million people [1], while 58 million suffer from hepatitis C virus (HCV) infection [2]. In a 2016 review, around 2,300,000 individuals were estimated to be co-infected with HIV/HCV, with more than half of these cases being among people who inject drugs (PWID) [3]. Evidence suggests that nationwide HCV elimination programs, typically coupled with the expansion of HCV testing and direct-acting antiviral (DAA) treatments, could mitigate new HCV infections among people with HIV (PWH) [4]. Greece has set as a national goal to achieve the Joint United Nations Programme on HIV/AIDS (UNAIDS) objectives of 95-95-95 with regard to HIV and national elimination with regard to HCV infections [4,5].

Treating HCV in co-infected PWH may have important effects on immune reconstitution. A multi-center collaboration within EuroCoord showed that acquiring HCV would result in a temporary decrease in CD4 counts even in chronically virally suppressed PWH [6]. Conversely, in already HCV-infected PWH, treatment with DAAs resulted in a substantial decrease in HIV DNA levels [7]. The low-grade inflammation maintained by HIV may have detrimental effects on liver health. The Italian Cohort Naive Antiretrovirals (ICONA) study showed that, in PWH, increased tumor necrosis factor-a (TNF-a) was associated with a 13-fold increased risk of having a fibrosis-4 (Fib-4) index score higher than 1.45, indicating advanced fibrosis and potentially cirrhosis [8].

Achieving sustained virological response (SVR) is important not just for the individual but also for the whole community, as it halts the further spread of HCV to other persons at risk, a strategy known as “treatment as prevention” [9].

A key component in any national plan to combat these infections is maintaining a constantly updated database of persons in need of treatment and of administered treatments’ outcomes [10,11]. Establishing a well-documented cascade of care (CoC) can assist in identifying areas that need improvement and targeting interventions to achieve the national goals. In HIV epidemiology, the CoC has been widely used to record diagnosis, linkage and retention to care, treatment, and virologic response [12]. Similarly, in HCV infection, the cascade involves diagnosis through antibodies, linkage to care, HCV RNA measurement, HCV treatment initiation, and proof of SVR [10].

Barriers to care for HIV/HCV co-infected individuals include active intravenous drug use, lack of socioeconomic stability (i.e., homelessness, lack of transportation, unemployment), mental health issues, and younger age [13,14]. In a large US cohort of mono- and co-infected individuals, the authors noted that people of color had greater difficulty accessing care, and women were also less likely to initiate HCV treatment even after three years of continuous HIV care [15].

In the broader Athens metropolitan area, Greece, an increasing number of HIV diagnoses amongst PWID was observed during 2011 [16,17]. Later on, based on emerging epidemiological data, it was realized that, in the same population, an HCV outbreak proceeded this HIV outbreak during 2008–2009. As a result, there was a significant increase in HIV/HCV co-infected individuals who required care for both HIV and HCV infections. In response to the 2011 HIV epidemic, the ARISTOTLE program (2012–2013) employed a “seek, test, treat, retain” intervention and succeeded in controlling the HIV epidemic [18]. Following the initial ARISTOTLE program, two other community-based programs, ARISTOTLE HCV-HIV (2018–2020) and ALEXANDROS (2019–2021), were implemented, aiming to increase the rates of diagnosis and linkage to care in HIV- and/or HCV-infected PWID living in Athens and Thessaloniki (second largest city in the north of Greece), respectively [19].

The goal of this study was to construct the CoC of HIV/HCV co-infected individuals and to identify potential areas of improvement in Greece.

## 2. Methods

We utilized data from the Athens Multicenter AIDS Cohort Study (AMACS), a collaborative, ongoing, population-based HIV cohort study in Greece. Currently, it encompasses 14 out of the 16 HIV clinics in Greece. A comprehensive description of the AMACS study is available elsewhere [20,21]. The study protocol underwent thorough scrutiny and received approval from multiple authoritative bodies, including the Hellenic Centre for Diseases Control and Prevention, the National Organization for Medicines (reference: NIS-23-01-05) on 5 June 2006, the Bioethics and Deontology Committee of the Medical School of the National and Kapodistrian University of Athens (reference: 319) on 18 October 2005, and the scientific committees of the respective hospitals associated with each participating clinic. The latest amendment of the protocol and informed consent forms were approved by the Bioethics and Deontology Committee of the Medical School of the National and Kapodistrian University of Athens (reference: 226) on 28 January 2020.

In this study, we employed data from PWH who tested positive for HCV according to anti-HCV tests (referred to from now on as HCV diagnosis) to construct an overall HIV/HCV CoC and assess differences by mode of HIV infection and migratory status, using data up to 31 December 2023. The present analysis was restricted to 11 of the 14 AMACS clinics that provided data for the whole spectrum of the HCV CoC. Initially, we presented basic demographic and clinical characteristics of the study population, overall, and by mode of HIV infection (infected with HIV through sex between men (MSM), people who inject drugs (PWID), infected with HIV through sex between men and women (MSW), and other or unknown modes of HIV infection). The factors examined included, among others, migratory status, age at HCV diagnosis, mortality before the date of universal access to DAAs for all HIV/HCV co-infected individuals in Greece (1 July 2017), CD4 at HIV diagnosis, and ART initiation. For categorical variables, absolute frequencies (N) and relative percentages (%) are provided, and between-group comparisons were conducted using Fisher’s exact tests (logrank for survival outcomes). In the case of continuous variables, the medians and interquartile ranges are presented, with corresponding comparisons were performed through Kruskal–Wallis tests.

In this study, we constructed a 7-stage CoC for HCV among PWH. The successive stages of the CoC were defined as follows:PWH with a positive anti-HCV test.Of those with anti-HCV(+), the percentage alive on 1 July 2017.Of those alive on 1 July 2017, the percentage retained in care (those disengaged from care were considered individuals who had never started DAAs and had not had a clinic visit up to two years before the clinic-specific database closure date).Of those in care, percentage with available data for chronic hepatitis C (CHC) infection (i.e., having initiated any treatment for HCV (DAAs or interferons) and/or having available HCV-RNA tests at/after a positive anti-HCV test and/or being positively genotyped).Among those with available data for CHC, the percentage of those with CHC. CHC was defined as initiating any treatment for HCV and/or having a positive HCV-RNA test and/or being positively genotyped. However, individuals who never initiated DAAs and were HCV-RNA-negative at their last HCV-RNA test after their anti-HCV positive test were assumed to have either experienced spontaneous clearance or been successfully treated with interferons. Thus, these individuals were not considered as having CHC.Of those with CHC, the percentage initiating DAAs.Among those initiating DAAs with available HCV-RNA tests at least 3 months after treatment completion, the percentage with SVR.

We estimated the proportions during the previous stage for stages 2–7. Subgroup analyses were conducted for PWID, MSM, MSW, and those with a migratory background.

We fitted multivariable logistic and Cox regression models to assess the between-group differences in the probability of passing to the next stage. Migratory status and mode of HIV infection were used as covariates in all analyses, irrespective of their statistical significance. More specifically, Cox regression models were fitted for the probability of dying from HIV diagnosis by 1 July 2017 before DAA initiation, and, among those alive, for the probability of being lost to follow-up prior to DAA initiation. In both models, age at HCV diagnosis and CD4 at HIV diagnosis were also included as covariates. Among individuals with CHC, in the logistic regression model for the probability of having initiated DAAs, the most recent CD4 count within 2 months prior to DAA initiation (last CD4 count for those that did not initiate DAAs), was also included as a covariate, indicating the immune system function, as did the percentage of time off ART from ART initiation to 2 months before DAA initiation (the latest follow-up date for those who did not initiate DAAs), as an indication of adherence to ART.

## 3. Results

### 3.1. Participants’ Characteristics

Among the 9059 PWH, 1213 individuals met the inclusion criteria (i.e., had a positive anti-HCV test). Most of them were PWID (N = 751, 61.9%), whereas 225 (18.5%) and 154 (12.7%) were MSM and MSW, respectively, and 83 (6.8%) had other or unknown modes of HIV infection. The basic demographic and clinical characteristics of the study population, overall, and by mode of HIV infection, are provided in Table 1. Of the 1213 individuals, 218 (18.0%) had a migratory background, which was most prevalent in MSW (N = 59 (38.3%), *p* < 0.001). The median (IQR) age at HCV diagnosis was 35.3 (29.8, 42.3) years, with PWID being the youngest group (33.6 years, *p* < 0.001). The median (IQR) CD4 count levels at HIV diagnosis were 327 (147–541) cells/μL, with the lowest levels observed among those with other or unknown mode of infection but also MSW and PWID had lower median levels than MSM (*p* < 0.001). Among those genotyped, the most prevalent HCV genotype was 3 followed by 1a, with significant differences among risk groups (genotype 3 being the most prevalent in PWID and 1a in MSM; *p* = 0.032). A substantial proportion of the study population had CD4 counts at HIV diagnosis below 200 cells/μL (31.4% overall, 27.8% in MSM, 30.3% in PWID, 34.4% in MSW, and 47.9% in those with other or unknown modes of infection). In total, 1081 (89.1%) individuals initiated ART over the follow-up period, with the highest percentage among MSM (96.4%) and lower percentages among PWID (87.5%) and those with other/unknown modes of HIV infection (81.9%).

### 3.2. Overall CoC and Subgroup CoC Analyses

The overall HCV/HIV CoC is graphically presented in Figure 1, with the subgroup analyses provided in Figure 2. Of the 1213 included individuals, 1101 (90.8%) survived until 1 July 2017, and therefore, before the availability of DAA treatment (Figure 1). The crude survival probabilities were roughly equal to 90% for PWID, MSM, and MSW and were slightly higher for the migrant population (93.6%, Figure 2). Among those alive, 795 (72.2%) remained in care, i.e., 306 disengaged while alive from care before the initiation of DAAs. The highest retention in care rate was among MSM (82.4%) and the lowest among the migrant population (58.3%, Figure 2). Of those retained in care, 630 (79.2%) had available data to determine whether they were chronically infected with HCV, which were roughly equal across subgroups (Figure 2), that is, of the 1101 alive in July 2017, only 57.2% (i.e., 72.2%*79.2%) had available data to determine their DAA eligibility. Overall, 490 of the 630 (77.8%) had CHC, which implies that 140 (22.2%) individuals either had spontaneous clearance or were previously successfully treated with interferon-based regimens. The proportion of those with CHC tended to be higher in PWID (84.7%) compared to MSM (66.2%) (Figure 2). Among the 490 CHC-positive individuals eligible for DAA treatment, 348 (71%) initiated DAAs. The DAA initiation rates were about 72% for PWID, MSM, and migrants, and substantially lower in MSW (57.7%) (Figure 2). Of those initiating DAAs, 19 (5.5%) had insufficient follow-up time to evaluate their SVR, whereas 126 (36.2%) had no available HCV-RNA at least 3 months after DAA treatment completion. Among the 203 (58.3%) individuals with available HCV-RNA tests at least 3 months after treatment completion, 174 (85.7%) achieved SVR. The SVR rates were slightly lower for PWID (82.9%, Figure 2) compared to the remaining groups. Overall, assuming a response rate among those with unavailable SVR data similar to the observed response rate, it is estimated that 60.9% of those with CHC, thus eligible for DAA therapy, achieved an SVR, the percentages being 60.3% in PWID, 63.9% in MSM, 57.7% in MSW, and 65.9% in migrants.

### 3.3. Mortality before DAA Treatment

In total, 112 (9.2%) of the study participants died before 1 July 2017 and DAA initiation. The survival probability (95% CI) at 10 and 15 years after HIV infection was 94.4% (91.2%, 97.7%) and 90.4% (85.8%, 95.2%) in MSM, 90.4% (87.9%, 92.9%) and 84.5% (79.3%, 90.1%) in PWID, and 91.1% (86.1%, 96.4%) and 87.6% (80.9%, 94.8%) in MSW, although the differences did not meet significant levels (*p* = 0.122). Table 2 presents the results from the corresponding multivariable Cox regression model. Notably, no significant differences were observed in mortality rates based on migratory status (*p* = 0.271). PWID exhibited a 91% higher mortality rate (95% CI: 14%, 222%) compared to MSM, with MSM showing the lowest mortality rate overall (Table 2). However, these differences were overall only marginally significant (*p* = 0.064). As anticipated, older age at HCV diagnosis appeared to be associated with increased mortality, although this relationship did not reach statistical significance (*p* = 0.136). Additionally, individuals with fewer than 200 CD4 cells/μL at HIV diagnosis were approximately twice as likely to die compared to those with more than 500 cells/μL at HIV diagnosis (overall *p* = 0.002).

### 3.4. Retention in Care before DAA Treatment

Table 2 also presents the results from the multivariable Cox regression models assessing the likelihood of loss to follow-up before DAA treatment initiation if alive on 1 July 2017. Migrants exhibited a twofold hazard of disengaging from care before DAA treatment (*p* < 0.001), while MSM had the lowest probability of loss to follow-up (overall *p* < 0.001). Specifically, PWID had a 135% (95% CI: 61%, 242%) higher hazard for loss to follow-up compared to MSM; individuals with fewer than 200 CD4 cells/μL at HIV diagnosis tended to have the lowest retention rate, although the overall significance did not meet the nominal level (*p* = 0.070). The effect of age at HCV diagnosis was insignificant (*p* = 0.743).

### 3.5. Availability of Data for CHC and CHC

The probability of having available data for determining the CHC status did not differ significantly by mode of HIV infection or migrant status. Among those with available data, the probability of having CHC did not differ by migrant status, whereas PWID had 183% (95% CI: 79%, 346%) higher odds of having CHC compared to MSM after adjusting for migrant status.

### 3.6. Initiation of DAAs

The median (IQR) time from CHC diagnosis (from 1 July 2017 if CHC diagnosis occurred before 1 July 2017) to DAA initiation, as estimated by the Kaplan–Meier estimator after excluding 34 (6.9%) individuals who initiated DAAs before 1 July 2017, was, overall, 0.8 (0.4, 5.8) years. Table 3 presents the results from the multivariable logistic regression model assessing the probability of DAA initiation among CHC-positive individuals. The DAA initiation percentages did not differ by migrant status (*p* = 0.223). After adjusting for the remaining factors in the model, MSW still exhibited 51% (95% CI: 77%, −6%) lower odds of initiating DAAs compared to MSM, with similar DAA initiation rates across the other risk groups (i.e., mode of HIV infection) but the overall significance level for mode of HIV infection did not reach the nominal level (*p* = 0.109). Individuals with a very low most recent CD4 count (<100 cells/μL) had substantially lower odds of initiating DAAs compared to other groups (*p* = 0.035). Notably, a 10% increase in the percentage of time spent off ART between ART initiation and 2 months before DAA initiation (maximum follow-up time for those never initiating DAA) was associated with 46% (95% CI: 50%, 20%) lower odds of initiating DAA treatment. Specifically, the average percentage of time off ART 2 months before DAA initiation was 5.2% for those who did not initiate DAA treatment, and 1.0% for those who did.

### 3.7. SVR

The results from the multivariable logistic model for the probability of SVR among those who initiated DAAs and had available HCV-RNA data at least three months after treatment completion showed that the SVR rates did not differ by migrant status (*p* = 0.575) or mode of HIV infection (*p* = 0.872), although there was some insignificant tendency for lower SVR rates among PWID (Table 4). The availability of data on SVR did not differ by mode of HIV infection (*p* = 0.906) or migratory status (*p* = 0.979).

## 4. Discussion

In this study, we have provided a representative picture of the state of HCV cure rates amongst HIV/HCV co-infected individuals in Greece. Our results reveal substantial gaps in the HCV CoC in HIV-HCV co-infected individuals, with the main barriers being low retention to HIV care rates and, for those remaining in care, delays in DAA initiation for various reasons; vulnerable populations, such as PWID and migrants, were the most affected.

The HIV/HCV joint epidemic, as expected, was driven mostly by PWID, with more than half of the participants belonging to this transmission group, but there is also a noteworthy percentage of MSM and MSW. It should be noted that the participants belonging to the latter group were less likely to have been offered HIV testing throughout their lifetime unless they presented with an indicator condition, as opposed to MSM, who would have tested preemptively under the assumption that they are at higher risk due to their sexual practices [22,23]. Amongst the transmission groups, migrants are overrepresented in the MSW group, with immigration status being another barrier to healthcare access. The lower CD4 counts observed in PWID and MSW were also an expected finding given the increased prevalence of late HIV presentation in these populations [23].

An unfortunate finding was that 9.2% of the participants died before having the opportunity to receive DAA treatment for HCV. This is in accordance with the findings reported in a recently published study, in which the crude mortality rates among PWID, recruited from the two community-based programs in the two largest Greek cities (Athens and Thessaloniki) during 2018–2022, were estimated [24]. It has been shown that treatment with DAAs in co-infected persons is associated with a 60–70% decrease in the mortality rate [25]. Presumably, those who had been treated with DAAs were those who were retained in care and received ART. In our cohort, PWID were more likely to die, presumably due to drug-related reasons, before DAAs became available compared to MSM. All-cause mortality in PWID in Athens has increased over the past five years, with daily opioid use and not being enrolled in an opioid-substitution program being identified, among others, as significant risk factors [24]. HCV/HIV co-infection has been linked to greater mortality compared to either mono-infection [24,26].

Migrants and PWID were less likely to be retained in care, reflecting again the inherent barriers to care faced by these populations. However, those overcoming this barrier would eventually receive treatments and be equally successful in achieving SVR compared to non-migrants and non-PWID. There exists a hierarchy of modifiable barriers to care in every healthcare system; system-level barriers (e.g., lack of providers, lack of patient navigation, lack of coordination among services), provider-level barriers (e.g., physician prejudice, referral bias, lack of knowledge) and patient-level barriers (e.g., adherence, substance use, socioeconomic problems) [27]. In a cohort from California, non-white co-infected individuals were significantly less likely to be referred for treatment whereas, on the other hand, persons with a history of alcohol or drug abuse were more likely to receive treatment [28].

People with lower CD4 counts and with greater time off ART were less likely to receive HCV treatment with DAAs. We do not have access to adherence data, but it can be hypothesized that individuals with greater recent CD4 counts and lesser time off ART are those with better adherence. The existing literature verifies our findings, as individuals with detectable viral loads and/or lower CD4 counts were less likely to receive treatment [28,29,30].

A large proportion of patients treated with DAAs was never evaluated for SVR. There is a gap of evidence in the existing literature as well as in our own data with regard to people who have not been evaluated for SVR. Brooks et al. reported a very high proportion of patients returning for SVR evaluation at 93.2% [29]. In European cohorts, about 90% of treated individuals returned for SVR evaluation, with large disparities between Eastern Europe compared to the rest of the continent [31].

Even though DAA treatments have documented success exceeding 90% in most circumstances, there is a paucity of real-world data for co-infected HCV/HIV individuals, and even fewer attempts to compare co-infected with mono-infected persons [32]. In our cohort, SVR was attained by 85.7% of participants, with relatively small variability among subgroups, but with PWID nonetheless having the lowest success rate. Among the participants eligible for treatment, that is, with CHC, we expect roughly 61% to have received DAAs and achieved SVR. In the available literature, this percentage varies greatly among countries. In certain cases, such as that in Belarus or a cohort from San Diego, the percentages of people cured were as low as 11.2% and 7%, respectively [28,31].

On the other hand, compared to those starting DAA treatment, the SVR rates in HIV/HCV co-infected individuals approach the rates seen in HCV mono-infected persons. In the HepCAUSAL a large multicohort collaboration of HIV/HCV co-infected individuals, 86% of the treated individuals had reported data for SVR; however, the observed overall SVR rate was as high as 93%, being similar across all investigated subgroups (age group, fibrosis stage, mode of HIV transmission, CD4 counts) [33]. In an Asia–Pacific cohort, even though only 40% of eligible people were treated, 90% of those treated achieved SVR [34]. In the American cohorts, SVR has been reported in 50% and 78% (44–100%) of participants in San Diego and Connecticut, respectively [28,29]. A study from Colombia found no differences in SVR rates between HIV/HCV co-infected and HCV mono-infected persons, with both groups attaining over 95% SVR [35]. A published Italian cohort with 55% HCV treatment-experienced participants and 67% with cirrhosis found no difference in SVR rates regardless of HIV status or underlying cirrhosis, where the only factor predicting failure to achieve SVR was being infected with HCV genotype 3 [36]. Finally, Carvalho et al. reported surprisingly even greater success in co-infected individuals compared to the mono-infected [37].

Our study, of course, is subject to some limitations. AMACS encompasses 14 out of the 16 HIV clinics in Greece. However, the two clinics that are not included account only for a small percentage of the PWH in care in Greece. Additionally, it has been shown that AMACS is representative of the broader HIV-diagnosed population in Greece [38]. In this analysis, 3 of the 14 AMACS clinics could not provide data on HCV treatment, as patients were administered for HCV treatment to other, mainly hepatological, clinics and, unfortunately, we had no access to these data at the time this analysis was conducted. Nevertheless, we have no reason to believe that this would bias our estimates, as similar HCV treatment guidelines are followed across all HIV clinics in Greece. Despite this, although AMACS may include a representative sample of HIV/HCV co-infected individuals living in Greece, one should not generalize our findings to different countries with different epidemiology and healthcare systems.

At different stages of the HCV CoC, a proportion of participants are lost to follow-up, leading to missing outcomes across the CoC stages. However, this is expected to some extent in any cohort study. A substantial percentage of those who received DAA treatment was never evaluated for SVR. This could be due to missing (unrecorded) HCV-RNA results or due to being lost to HIV follow-up. To partly address this issue, we compared the percentages of missing SVR data across risk groups; we found no significant differences in missing data rates, suggesting that our estimates of the SVR rates are unlikely to be seriously affected by selection bias. In our study, 142 of the 490 individuals with CHC did not initiate DAAs. Although we do not have access to specific data on the reasons for not initiating DAAs, we have identified several factors associated with the probability of initiating DAAs. Furthermore, even though we have explored some potential predictors of SVR, we acknowledge that they do not capture all possible reasons for treatment failure. Adherence to DAA treatment is an essential factor affecting treatment outcomes. Unfortunately, we do not have access to data on adherence or other potential predictive factors for initiating DAAs and treatment failure, such as socioeconomic factors. However, a detailed investigation of factors predicting access to care and/or SVR failure was out of the scope of this analysis, mainly due to not having access to relevant data.

## 5. Conclusions

SVR amongst HCV/HIV co-infected individuals can be achieved in proportions similar to that in mono-infected persons. Overcoming barriers to healthcare is imperative in order to achieve national goals, eliminate HCV infection, and provide better overall healthcare in those vulnerable populations that need it most.

## Figures and Tables

**Figure 1 viruses-16-00885-f001:**
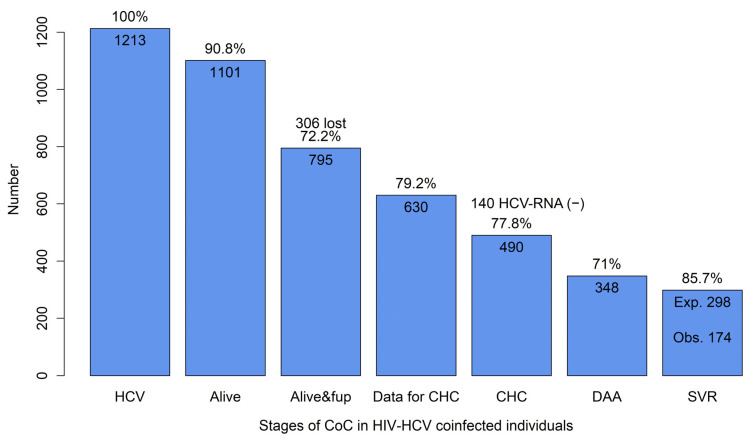
Stages of cascade of care in HIV/HCV co-infected individuals. All percentages were evaluated with respect to the number of individuals in the previous stage. HCV: hepatitis C virus; CHC: chronic hepatitis C; DAA: direct-acting antivirals; SVR: sustained virological response; Exp.: expected; Obs.: observed; CoC: cascade of care.

**Figure 2 viruses-16-00885-f002:**
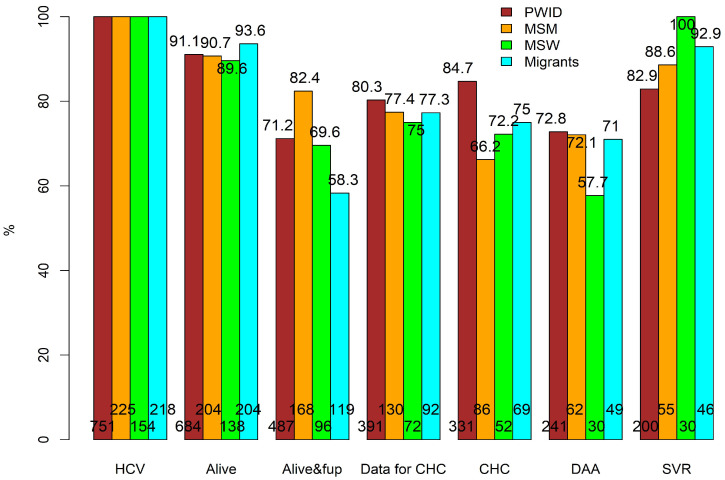
Stages of cascade of care in HIV/HCV co-infected by risk group: PWID in brown, MSM in orange, MSW in green, and migrants in cyan. All percentages were evaluated with respect to the number of individuals in the previous stage. The respective numbers for each group are presented at the bottom of each bar. PWID: people who inject drugs; MSM: men who have sex with men; MSW: sex between men and women; HCV: hepatitis C virus; fup: under follow-up; CHC: chronic hepatitis C; DAA: direct-acting antivirals; SVR: sustained virological response.

**Table 1 viruses-16-00885-t001:** Demographic and clinical characteristics of the HCV/HIV co-infected individuals.

	MSM (N = 225; 18.5%)	PWID (N = 751; 61.9%)	MSW (N = 154; 12.7%)	Other/Unknown (N = 83; 6.8%)	Total (N = 1213)	*p* Value
Migrant status						<0.001
No	203 (90.2%)	635 (84.6%)	95 (61.7%)	62 (74.7%)	995 (82.0%)	
Yes	22 (9.8%)	116 (15.4%)	59 (38.3%)	21 (25.3%)	218 (18.0%)	
Age at HCV diagnosis (years)						<0.001
Median (Q1, Q3)	38.3 (32.3, 46.0)	33.6 (28.9, 39.6)	38.4 (31.5, 46.5)	37.7 (29.5, 44.7)	35.3 (29.8, 42.3)	
CD4 at HIV diagnosis (cells/μL)						<0.001
N-Miss	2	18	3	10	33	
Median (Q1, Q3)	387.0 (171.0, 582.5)	328.0 (164.0, 556.0)	319.0 (112.5, 482.5)	224.0 (79.0, 384.0)	327.0 (147.0, 540.5)	
ART initiation						<0.001
No	8 (3.6%)	94 (12.5%)	15 (9.7%)	15 (18.1%)	132 (10.9%)	
Yes	217 (96.4%)	657 (87.5%)	139 (90.3%)	68 (81.9%)	1081 (89.1%)	
HCV genotype ^1^						0.032
N-Miss	56	179	35	14	284	
1a	12 (40.0%)	47 (30.9%)	3 (17.6%)	2 (28.6%)	64 (31.1%)	
1b	2 (6.7%)	6 (3.9%)	5 (29.4%)	2 (28.6%)	15 (7.3%)	
3	12 (40.0%)	75 (49.3%)	7 (41.2%)	3 (42.9%)	97 (47.1%)	
4	4 (13.3%)	24 (15.8%)	2 (11.8%)	0 (0.0%)	30 (14.6%)	
^1^ Among those chronically infected with HCV						

MSM: men who have sex with men; PWID: people who inject drugs; MSW: sex between men and women; HCV: hepatitis C virus; HIV: human immunodeficiency virus; ART: anti-retroviral treatment.

**Table 2 viruses-16-00885-t002:** Factors affecting the hazard of dying prior to 1 July 2017 or DAA initiation and the hazard of being lost to follow-up if alive on 1 July 2017 and prior to initiating DAAs. Results are from multivariable Cox regression models.

	A. Mortality Model (N = 1213)			B. Loss-to-Follow-up Model (N = 1101)		
Characteristic	HR ^1^	95% CI ^1^	*p* ^2^	HR ^1^	95% CI ^1^	*p* ^2^
Migrants	0.71	0.38, 1.31	0.271	1.98	1.51, 2.60	<0.001
Mode of HIV infection			0.064			<0.001
MSM	—	—		—	—	
PWID	1.91	1.14, 3.22	0.015	2.35	1.61, 3.42	<0.001
MSW	1.54	0.79, 3.04	0.208	1.83	1.16, 2.91	0.010
Other/Unknown	1.01	0.44, 2.30	0.978	1.70	0.98, 2.96	0.060
Age at HCV diagnosis (per 10 years)	1.15	0.96, 1.37	0.136	0.98	0.87, 1.11	0.743
CD4 at HIV diagnosis (cells/μL)			0.002			0.070
(0, 200)	—	—		—	—	
(200, 350)	0.60	0.37, 0.98	0.043	0.64	0.46, 0.90	0.011
(350, 500)	0.33	0.17, 0.63	<0.001	0.76	0.54, 1.07	0.114
>500	0.53	0.32, 0.86	0.011	0.82	0.61, 1.11	0.199

^1^ HR = hazard ratio; CI = confidence interval. ^2^ Global *p*-values also presented. HIV: Human immunodeficiency virus; MSM: men who have sex with men; PWID: people who inject drugs; MSW: sex between men and women; HCV: hepatitis C virus.

**Table 3 viruses-16-00885-t003:** Results from a multivariable logistic regression model for the probability of having initiated DAAs among individuals with chronic HCV in relation to migratory status, mode of HIV infection, most recent CD4 count before DAA initiation, and percentage of time off ART before DAA treatment.

Characteristic	OR ^1^	95% CI ^1^	*p* ^2^
Migrants	1.49	0.80, 2.88	0.223
Mode of HIV infection			0.109
MSM	—	—	
PWID	1.10	0.61, 1.92	0.748
MSW	0.49	0.23, 1.06	0.071
Other/Unknown	1.15	0.37, 4.09	0.820
Most recent CD4 before DAA treatment (cells/μL)			0.035
(0, 100)	—	—	
(100, 350)	3.50	1.35, 9.15	0.010
>350	2.51	1.04, 6.02	0.038
% of time off ART before DAA (per 10%)	0.64	0.50, 0.80	<0.001

^1^ OR = odds ratio; CI = confidence interval. ^2^ Global *p*-values also presented. HIV: Human immunodeficiency virus; MSM: men who have sex with men; PWID: people who inject drugs; MSW: sex between men and women; DAA: direct-acting antiviral; ART: antiretroviral treatment.

**Table 4 viruses-16-00885-t004:** Results from a multivariable logistic regression model for the probability of achieving SVR among those initiating DAA treatment with available HCV-RNA data at least 3 months after DAA treatment completion in relation to migratory status and mode of HIV infection.

Characteristic	OR ^1^	95% CI ^1^	*p* ^2^
Migrants	1.56	0.40, 10.3	0.575
Mode of HIV infection			0.872
MSM	—	—	
PWID	0.66	0.18, 1.88	0.468
MSW	NA ^3^		0.987
Other/Unknown	1.12	0.14, 23.4	0.926

^1^ OR = odds ratio; CI = confidence interval. ^2^ Global *p*-values also presented. ^3^ Could not be estimated due to 100% SVR probability among MSW. HIV: Human immunodeficiency virus; MSM: men who have sex with men; PWID: people who inject drugs; MSW: sex between men and women; DAA: direct-acting antiviral; ART: antiretroviral treatment.

## Data Availability

AMACS individual-level data were derived from collaborating HIV clinics, and although the individual data do not include patient names or identifying information of the participants, as the data contain potentially sensitive information, there are ethical restrictions imposed by the Ethics Committee of the Medical School of the National and Kapodistrian University of Athens (NKUA). Anonymized individual data can be shared after interested researchers submit a concept sheet to the AMACS steering committee (chair: Giota Touloumi, email: gtouloum@med.uoa.gr) and the Ethics Committee of Medical School of NKUA (bioethics@med.uoa.gr).

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
