# Peer review of "HCV Cascade of Care in HIV/HCV Co-Infected Individuals: Missed Opportunities for Micro-Elimination"

_viruses, 2024, doi:10.3390/v16060885_

Round 1

Reviewer 1 Report

Comments and Suggestions for Authors

Thomadakis et al. evaluated an HCV “Cascade of care” for HIV/HCV co-infected individuals and tried to identify potential areas of improvement. Despite the interesting issue there are several concerns.

1) The cohort to consider for all the analysis is composed of 795 patients alive on follow-up and the subsequent rates of the different steps of the cascade should have this number as the denominator 

2) 11 among the 16 HIV clinics in Greece provided data for the spectrum of HCV cascade. This could be a major limitation of the study that need to be discussed

3) One interesting aspect is to better understand why 142 of the 490 CHC positive patients didn’t initiate DAAs as well as to known the predictors of non response. Any data on adherence of DAAs could be added

4) HCV genotype is not available, and this could further help stratify the epidemiology of HIV/HCV co-infection in Greece.

5) Were the 306 disengaged patients just lost at follow-up or were there any cases of death after July 1, 2017? Which subgroup did they belong to?

6) In conclusion is it stated that SVR among HIV/HCV co-infected patients can be achieved in proportions similar to HCV mono-infected patients. However, in a sort of "ITT analysis", that considered missing value as a failure,  nearly 50% of those treated didn't achieve SVR.

Reviewer 2 Report

Comments and Suggestions for Authors

The study by Thomadakis et al., entitled “HCV Cascade of Care in HIV/HCV Coinfected Individuals: Missed Opportunities for Micro-Elimination” aimed to construct the cascade of care (CoC) of HIV/HCV co-infected individuals and identified potential areas of improvement in Greece. This is an interesting study, and findings may be useful for the management of HIV/HCV patients.

However, I have following major concerns about the study.

1)      Manuscript is poorly written with several grammatical errors and sometimes it is hard to follow the article. Manuscript needs to rewrite for better understanding and clarity. Authors need to give the full form of all the acronyms used in the manuscript for example UNAIDS, ICONA, Fib-4 etc.

2)      Authors need to mention full form of all the first time used acronyms in the abstract of the manuscript. First give the full form and then mention the acronyms. In the present form it is hard to understand the abstract and need to refer the other portion of the manuscript to understand it.

3)      Authors need to define the clinical terminology used throughout the manuscript, for example Sustained virological response (SVR).

Comments on the Quality of English Language

Manuscript is poorly written with several grammatical errors and sometimes it is hard to follow the article. Manuscript needs to rewrite for better understanding and clarity.
